# Dynamic Changes in Plant Resource Use Efficiencies and Their Primary Influence Mechanisms in a Typical Desert Shrub Community

Yan Jiang [1,2], Yun Tian [1,2,*], Tianshan Zha [1,2], Xin Jia [1,2], Charles P.-A. Bourque [3], Peng Liu [1,2], Chuan Jin [1,2], Xiaoyan Jiang [1,2], Xinhao Li [1,2], Ningning Wei [1,2] and Shengjie Gao [1,2]

[1] School of Soil and Water Conservation, Beijing Forestry University, Beijing 100083, China; jiangyansmile88@163.com (Y.J.); tianshanzha@bjfu.edu.cn (T.Z.); xinjia@bjfu.edu.cn (X.J.); spiritlover@126.com (P.L.); jinchuan@bjfu.edu.cn (C.J.); jiangxiaoyan916@163.com (X.J.); xinhaoli@bjfu.edu.cn (X.L.); wei1196020090@163.com (N.W.); shengjiegao127@126.com (S.G.)

[2] Beijing Engineering Research Center of Soil and Water Conservation, Beijing Forestry University, Beijing 100083, China

[3] Faculty of Forestry and Environmental Management, 28 Dineen Drive, University of New Brunswick, Fredericton, NB E3B 5A3, Canada; cbourque@unb.ca

* Correspondence: tianyun@bjfu.edu.cn

**Abstract:** Understanding plant resource use efficiencies (*RUE*s) and their tradeoffs in a desert shrub community, particularly as it concerns the usage of water, light, and nitrogen, remains an ecological imperative. Plant *RUE*s have been widely used as indicators to understand plant acclimation processes to unfavorable environmental conditions. This study aimed to examine seasonal dynamics in *RUE*s in two widely distributed plant species in a typical desert shrub community (i.e., *Artemisia ordosica* and *Leymus secalinus*) based on in-situ measurements of leaf photosynthesis, specific leaf area (*SLA*), leaf nitrogen concentration (i.e., $N_{mass}$ + $N_{area}$), and several site-related abiotic factors. Both species exhibited significant seasonal variation in *RUE*s, with a coefficient of variation (CV) > 30% and seasonal divergence among the various *RUE*s. Seasonal divergence was largely controlled by variation in stomatal conductance (*Gs*), which was in turn influenced by variation in soil water content (*SWC*) and water vapor pressure deficit (*VPD*). *RUE*s between species converged, being positively correlated, yielding: (i) $r^2 = 0.40$ and $p < 0.01$ for *WUE*; (ii) $r^2 = 0.18$ and $p < 0.01$ for *LUE*；and (iii) $r^2 = 0.25$ and $p < 0.01$ for *NUE*. *RUE*s for *A. ordosica* were mostly larger than those for *L. secalinus*, but less reactive to drought. This suggests *A. ordosica* was more conservative in its usage of available resources and was, therefore, better able to adapt to arid conditions. Resource use strategies between species differed in response to drought. Desert shrubs are projected to eventually replace grasses, as drought severity and duration increase with sustained regional climate change.

**Keywords:** dryland; *Artemisia ordosica*; *Leymus secalinus*; resource use efficiency; light use efficiency; water use efficiency; nitrogen use efficiency; tradeoffs

## 1. Introduction

Drylands (arid and semiarid areas) cover 39% of the earth's land surface and are home to about 20% of the world's human population [1]. Ecosystems in drylands are highly vulnerable to global environmental change and desertification [2]. Given the speed and intensity of climate change and socioeconomic development that risk aggravating environmental and socioeconomic problems on various spatial scales (e.g., land degradation, poverty, declining food and water security), research on both social and ecological system processes, as well as their interactions, is urgently needed [3,4]. Recent studies highlighting the importance of semiarid ecosystems in their contribution to terrestrial net

primary production are also key to forming meaningful sustainable development goals (SDGs) for drylands [1,4–6].

Plants in drylands generally experience more environmental stress, especially as severity and duration of droughts continue to increase under a warming climatic regime [7–9]. Consequently, it is expected that there will be changes in the supply of available plant resources, such as water ($W$), light ($L$), and nitrogen ($N$) for photosynthetic assimilation, as well as for the maintenance of functional plant structures. This may lead to changes in plant resource use efficiencies (i.e., *RUE*s, or individually through *WUE*, *LUE*, and *NUE*). Plant *RUE* (and its components) is defined as the ratio of photosynthetic assimilation to the consumption and subsequent usage of resources [10,11]. Photosynthesis is generally limited by $W$, $L$, and $N$, although other abiotic factors, such as temperature ($T$), can also play a crucial part, particularly at seasonal timescales [12]. The survival and growth of plants could be influenced by *RUE*s, which could ultimately cause fluctuations in ecological functioning. Plant *RUE*s have been widely used to understand plant acclimation processes to unfavorable environmental conditions expected to accompany climate change and associated extreme weather events.

Plant resources may change in unison. Changes in one resource will likely induce changes in other resource uses at the leaf, species, and community level, conditional on the temporal scale [13]. Changes in the availability of one resource may result in changes, not only in the use efficiency of that resource, but also in the use efficiencies of other resources [10,14]. For instance, water supply increases *NUE*, but decreases *WUE*, whereas the addition of $N$ slightly increases *WUE* at the expense of *NUE* at the leaf level. This suggests that the shift in the availability of one resource could engender disproportionate constraints on the use efficiencies of other plant resources [15]. Large transpiration rates and high solar irradiation proceed to maximize instantaneous photosynthesis, which correlates positively with *NUE* and negatively with both *WUE* and *LUE* [16]. Consequently, the individual response in *WUE*, *LUE*, and *NUE* to changes in environmental conditions are not self-governing. Therefore, plants may exhibit divergent tradeoffs between the various *RUE*s during acclimatization to prevailing site conditions. To better model plant responses to climate change in arid and semiarid areas, relative changes (tradeoffs) among *WUE*, *LUE*, and *NUE* and their causes need to be quantified.

*RUE*s and their response to environmental change have received considerable interest (e.g., [17–20]). Changing environmental factors constrain the variation in plant *RUE*s, as the factors influence the supply and demand of resources [20,21]. *RUE*s integrated over monthly timescales (i.e., *RUE*m) vary seasonally in response to variation in resource availability and weather conditions [16]. Crops and native vegetation that are adapted to water-limited conditions, achieve adaptation mainly by dehydration avoidance and escape, rather than having an innate ability to function in a dehydrated state or be desiccation tolerant [22]. Fischer and Maurer have shown that agricultural cultivars yield biomass under drought as a function of yield potential (yield without drought), drought susceptibility index, and severity of drought [23]. Improved *WUE* is usually expressed in an improved yield under water-limited conditions, only when there is a need to balance crop water use against a limited and known soil water supply [22]. Plant breeders and physiologists have long been concerned with drought resistance in plants. Just how dryland plants adjust their *RUE*s to resist drought remains uncertain.

Species is by far the most influential factor explaining the variance in leaf photosynthetic assimilation [24]. *RUE*s, key plant function, acclimate to environmental fluctuations through coordination and tradeoffs among plant morphological, physiological, and biochemical traits in achieving optimal resource usage and adaptation to prevailing environmental conditions [18,25]. Plant production in water-limited environments is very often affected by constitutive plant traits that allow maintenance of a high plant water status [22]. Owing to the likely diversity of factors and factor interactions to *RUE*s, dynamic changes in *RUE*s and their primary influence pathways or mechanisms are not entirely understood, and thus further studies are needed, especially for dryland ecosystems.

Plant ecosystem health in arid and semiarid areas is rapidly deteriorating as water shortages become more frequent [26]. Responses of plant communities to environmental change are typically regulated by the dominant species in the community [27,28]. It has been reported that the neighborhood of a tree can have a significant impact on functional traits involved in resource use [29]. Shrub-dominated plant communities in drylands of northwest China are commonly associated with the presence of *Artemisia ordosica* Krasch. (shrub) and *Leymus secalinus* (Georgi) Tzvel. (desert grass) [19]. These communities are widespread throughout the semiarid regions of the Mu Us Desert [30]. Their sustainability and presence, however, are being greatly affected by severe limitations in available soil water emergent in many parts of the desert complex [26,31].

This study aims to understand the characteristics and influence mechanisms and pathways responsible for the observed variation in *RUE*s in both *A. ordosica* and *L. secalinus* in response to dry conditions. The specific objectives of the study are to: (i) examine seasonal dynamics in the individual components of *RUE*s (i.e., *WUE*, *LUE*, and *NUE*); (ii) determine whether a level of convergence exists in *RUE*s between the two plant species; and (iii) clarify the role of environmental factors in the control of seasonal dynamics in *RUE*s. The study will have important implications for clarifying the acclimatization capacity of dryland shrub-grass associations and understanding how drylands respond to ongoing environmental change. It is a key theme of SDGs of drylands, regarding their social-ecological system dynamics and drivers [4,32]. The study has the potential to deliver both novel scientific insights and development impact consistent with the aspirations of United Nations' SDGs for 2030.

## 2. Methods

### 2.1. Site Description

The study site is located at the Yanchi Research Station of Beijing Forest University (37°42′31″ N, 107°13′47″ E, 1530 m above mean sea level), Ningxia, northwest China (Figure 1). The site is representative of a transitional zone between arid and semiarid conditions at the southern edge of the Mu Us Desert. The prevailing regional climate is temperate arid and semiarid, characterized with scarce rainfall, irregularly distributed, and variable from year to year. The mean annual temperature (based on 1954–2020 data) is 8.4 °C, and the mean annual precipitation is 293 mm. Most precipitation (>70%) occurs during June–September (data source: Yanchi Meteorological Station, Yanchi Research Station).

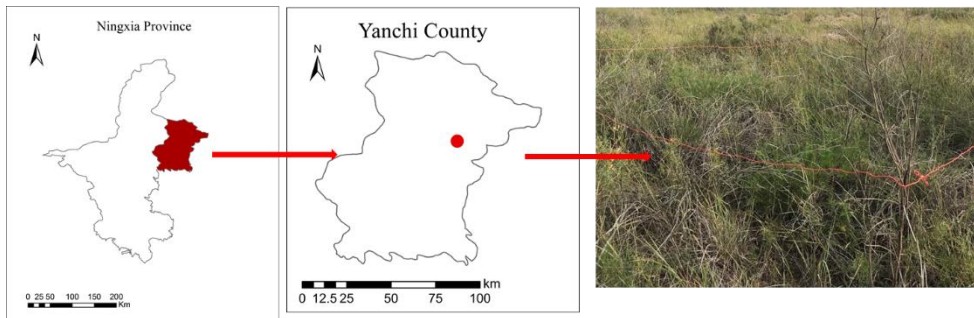

**Figure 1.** Map of the study site and its dominant plant cover.

The soil at the site involves sand, with a bulk density of $1.54 \pm 0.08$ g cm$^{-3}$, a total porosity of $35.70 \pm 3.83\%$, a field capacity of $20.31 \pm 3.33\%$ (g g$^{-1}$ × 100), and a permanent wilting point of $3.64 \pm 0.37\%$ in the upper 10 cm of the soil profile (g g$^{-1}$ × 100; mean ± standard deviation, $n = 16$). The landscape of this region is a typical inland dune ecosystem with very distinct habitat types [33]. The shrub cover at the site consists mostly of *A. ordosica*, and smaller amounts of *Hedysarum mongolicum* Turez.and *Salix psammophila* C. The most abundant herbaceous species include *L. secalinus*, S*tipa glareosa* P. Smirn., *Pennisetum*

*centrasiaticum* Tzvel., and *Setaria viridis* (L.) Beauv.. The main canopy of the community is about 1 to 1.5 m tall. Shrub roots are distributed mainly in the upper 10–50 cm of the soil.

### 2.2. Photosynthesis Gas-Exchange Measurements

The measurements were taken from plants enclosed within three 5 × 5 m² plots. Three individuals of both *A. ordosica* and *L. secalinus* were randomly selected in each plot ($n = 9$) assigned for in situ measurements [26,34,35]. Photosynthesis was measured on fully developed leaves on the south-facing side of each assigned plant individuals, every 10 days from May to September 2019. The measurements were taken with a portable LI-6400 photosynthesis system, equipped with 2×3 cm² transparent light source chamber (Li-Cor Inc., Lincoln, NE, USA). Five measurements were taken from each individual plant, during each measurement period. All leaf gas-exchange characteristics were measured six times every ~2-h from 8:00–18:00 Local Beijing Time (Greenwich Mean Time + 8 h). Rates of photosynthesis and transpiration (*Pn* and *E*, in μmol m$^{-2}$ s$^{-1}$ and mmol m$^{-2}$ s$^{-1}$, respectively) and stomatal conductance (*Gs*, mol m$^{-2}$ s$^{-1}$) were measured, which were expressed per unit leaf area. The leaf area of fresh leaves was measured at the same time using an LI-3100C leaf area meter (LI-COR Environmental, Lincoln, NE, USA).

### 2.3. Measurement of Biotic Factors

Two samples of 10 leaves were collected immediately after each gas-exchange measurement on nearby plants with similar characteristics to the target plants, for both leaf area measurement and *N* content. After each leaf area measurement, fresh weight of sampled foliage was measured with an electronic balance. The fresh leaves were then oven-dried at 75 °C for 48-h for dry weight determination. The specific leaf area (*SLA*, cm² g$^{-1}$) was then calculated as the ratio of fresh leaf area to dry weight. The leaf *N* concentration was determined both on a dry weight and leaf area basis (i.e., $N_{mass}$ and $N_{area}$, g kg$^{-1}$ and g m$^{-2}$, respectively) using the Kjeldahl method [10].

### 2.4. Measurement of Abiotic Factors

All meteorological variables were measured with sensors installed on 6-m tall eddy-covariance tower, assembled next to the sampling area. Tower-based measurements included: (i) air temperature (*T*, °C) and relative humidity (*RH*, %) taken with a thermo-hygrometer (HMP155A, Vaisala, Finland); (ii) net radiation (*Rn*, W m$^{-2}$) with a CNR-4 sensor (Kipp & Zonen, Netherlands); and (iii) incident photosynthetically active radiation (*PAR*, μmol m$^{-2}$ s$^{-1}$) with a quantum sensor (PAR-LITE, Kipp & Zonen, The Netherlands). Half-hourly soil water content (*SWC*, m³ m$^{-3}$) was monitored at a 30-cm depth within a 10-m radius around the tower, with three replicate sensors (ECH2O-5TE, Decagon Devices, Pullman, WA, USA). Rainfall (*PPT*, mm) was measured using a tipping bucket raingauge (TE525WS, Campbell Scientific Inc., Logan, UT, USA) set in an opening approximately 50 m from the tower. Vapor pressure deficit (*VPD*, kPa) was calculated from tower-based measurements of *RH* and *T*.

### 2.5. Data Processing and Analysis

Water, light, and nitrogen use efficiencies (i.e., *WUE* in μmol mmol$^{-1}$, *LUE* in mol mol$^{-1}$, and *NUE* in μmol g$^{-1}$ s$^{-1}$, respectively) were calculated individually with the equations (1)–(3):

$$WUE = \frac{Pn}{E} \tag{1}$$

$$LUE = \frac{Pn}{PAR} \tag{2}$$

$$NUE = \frac{Pn}{N_{area}} \tag{3}$$

where $Pn$ is the net photosynthesis, $E$ the transpiration, $PAR$ the leaf surface photosynthetically active radiation, and $N_{area}$ the amount of $N$ per unit leaf area. Monthly water, light, and nitrogen use efficiencies were subsequently calculated as averages of daily $WUE$, $LUE$, and $NUE$. Major terms and their acronyms appear in Table A1.

### 2.6. Statistical Analysis

The coefficient of variation (CV) was used to quantify the seasonal variation in $RUE$s (i.e., $WUE$, $LUE$, and $NUE$) and biotic factors. To analyze relationships among $WUE$, $LUE$, and $NUE$ for a given species and to determine whether a level of convergence exists in $RUE$s between the two plant species, a standard major axis (SMA) operation was performed using *sma*, a procedure available in the *smatr* R-package. Paired student t-tests were performed in pairwise comparisons of $WUE$, $LUE$, and $NUE$ for *A. ordosica* and *L. secalinus*. To clarify the role of factors in the control of seasonal dynamics in $RUE$s, stepwise regression was used to find the most critical biophysical factors responsible for controlling $RUE$s. In addition, structural equation models (SEM) were used to assess the direct and indirect contributions of biotic and abiotic factors to variations in $RUE$s. The significance level was set at $p = 0.05$.

Drought days in 2019 were defined as those days with daily mean $SWC < 0.1$ m³ m⁻³ [9,26,36]. The average dates for the onset and end of the growing season were set at early May and end of September. Spring, summer, and autumn were defined as occurring in May, June–August, and September, respectively. All statistical analyses were performed in R-Studio ver. 3.6.3 (The R Development Core Team) and Origin2017 (OriginLab, Northampton, MA, USA).

### 3. Results

### 3.1. Variations in Biophysical Factors and Photosynthetic Parameters

Seasonal patterns in $PAR$ and $R$n were similar (Figure 2C). The range of daily means for $PAR$, $R$n, $VPD$, $T$, and $SWC$ during the growing season were 61.32–608.51 μmol m⁻² s⁻¹, 18.07–223.96 W m⁻², 0.08–2.23 kPa, 7.20–24.51 °C, and 0.04–0.12 m³ m⁻³, respectively (Figure 2). Peaks in $SWC$ corresponded with intermittent rain pulses. Low $SWC$ occurred during non-rainy days (Figure 2D). Drought was observed whenever $SWC < 0.10$ m³ m⁻³ (Figure 2D).

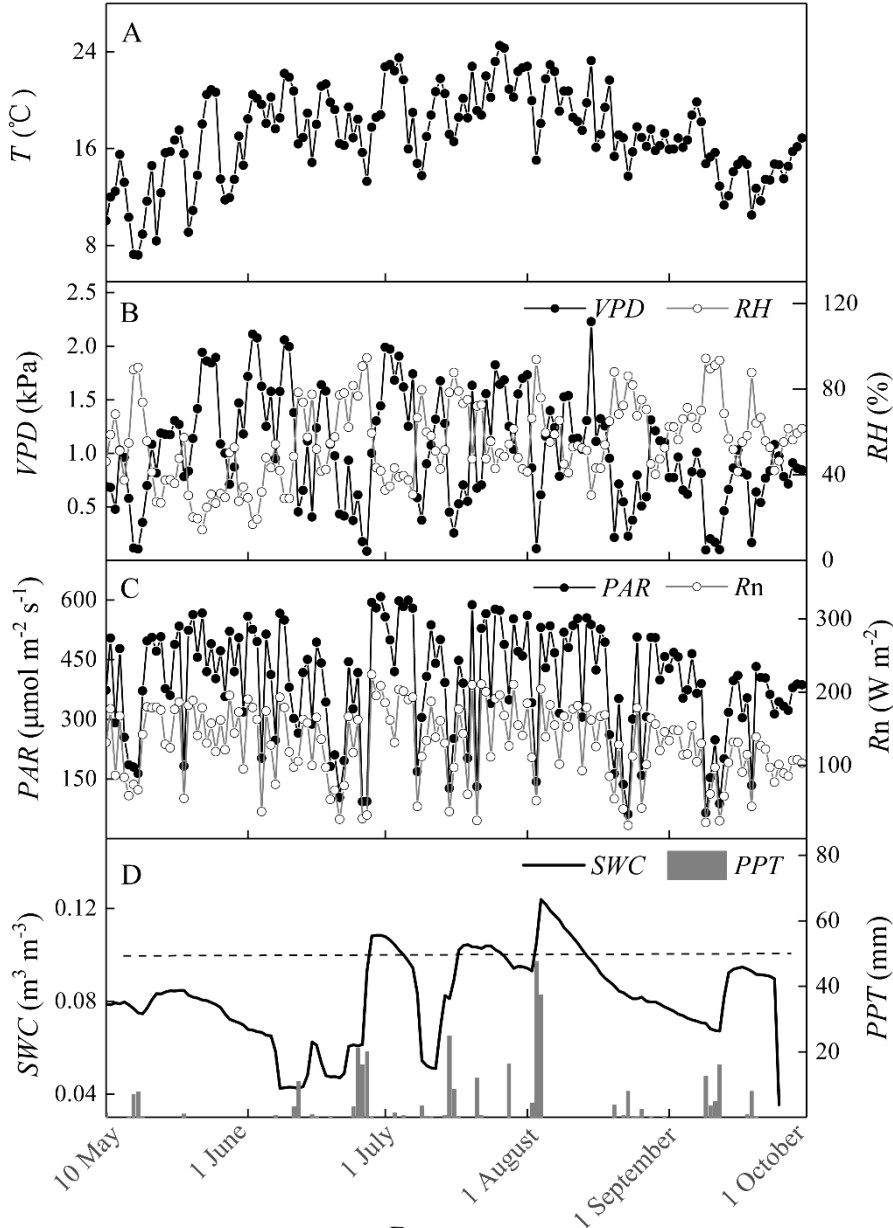

**Figure 2.** Temporal variation in daily mean air temperature (*T*, (**A**)), relative humidity (*RH*) and water vapor pressure deficit (*VPD*, (**B**)), net radiation (*R*n) and incident photosynthetically active radiation (*PAR*, (**C**)), and soil water content (*SWC*) and daily total precipitation (*PPT*, (**D**)) during a five-month field campaign from 1 May−30 September 2019. The horizontal dashed line in subfigure D represents the 0.10 m³ m⁻³ threshold assigned for *SWC*.

The two plant species had near-similar diurnal and seasonal tracking of $P$n, $E$, and $G$s (Figure 3). During the growing season, $P$n, $E$, and $G$s ranged from 6.12–23.55 μmol m⁻² s⁻¹, 0.13–0.71 mmol m⁻² s⁻¹, and 3.56–15.66 mol m⁻² s⁻¹, respectively, for *A. ordosica*, and 4.75–11.16 μmol m⁻² s⁻¹, 0.11–0.40 mmol m⁻² s⁻¹, and 3.06–9.55 mol m⁻² s⁻¹ for *L. secalinus* (Figure 3A–C). Values of photosynthetic parameters for *A. ordosica* were generally greater than those for *L. secalinus.* Plant parameters of *SLA*, $N$mass, and $N$area for *A. ordosica* varied less than those of *L. secalinus* over the same period, with CV of 16, 10, and 20% for *A. ordosica*, and 24, 12, and 34% for *L. secalinus* (Figure 4). Leaf $N$ concentrations (with respect to both $N$mass and $N$area) were lower for *A. ordosica* than those for *L. secalinus* (Figure 4B,C). *SLA* exhibited an opposite trend (Figure 4A), ranging between 59.01–126.88 and 38.54–94.82 cm g⁻¹ for *A. ordosica* and *L. secalinus*, respectively.

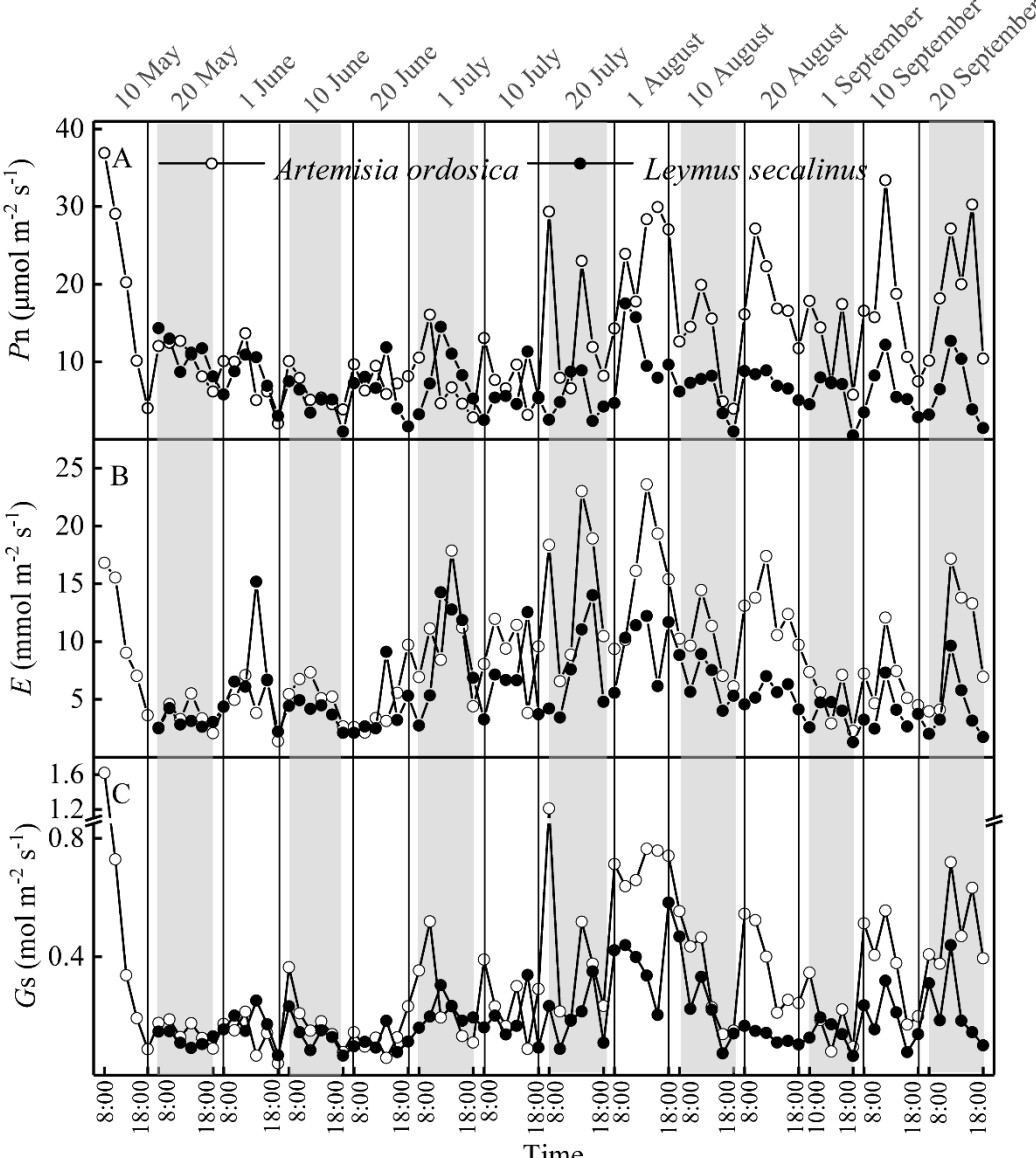

**Figure 3.** Diurnal variation in rates of photosynthesis (*P*n, (**A**)), transpiration (*E*, (**B**)), and stomatal conductance (*G*s, (**C**)) for *A. ordosica* and *L. secalinus* during a five-month field campaign from 1 May–30 September 2019. Photosynthetic parameters for *L. secalinus* are missing during the 1–10 May period.

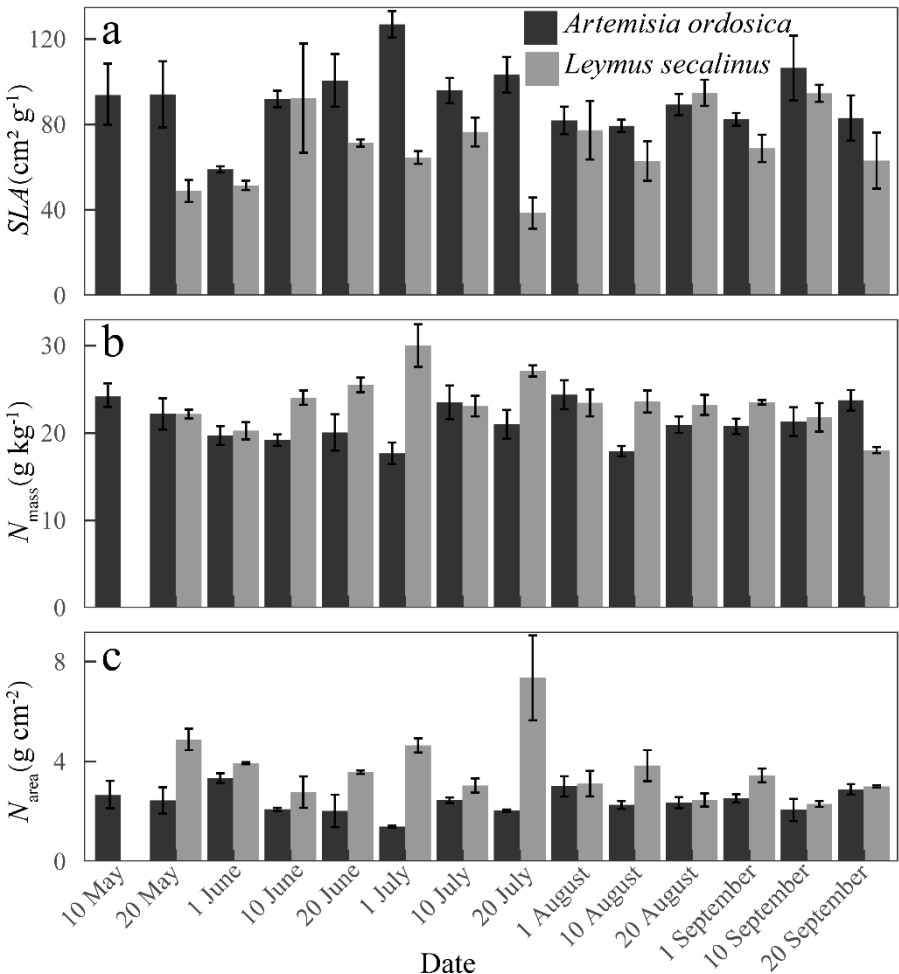

**Figure 4.** Leaf characteristics, including specific leaf area (*SLA*, (**A**)), leaf *N* concentration by dry weigh (*N*mass, (**B**)), and *N* concentration per unit leaf area (*N*area, (**C**)) for *A. ordosica* and *L. secalinus* during the five-month field campaign from 1 May−30 September 2019.

### 3.2. Seasonal Dynamics in Plant Resource Use Efficiencies

Both species had similar seasonal patterns in *RUE*s (i.e., *WUE*, *LUE*, and *NUE*) from May–September 2019 (Figures 5 and 6). *RUE*s in *A. ordosica* were mostly larger than those in *L. secalinus*, with mean values of 1.84 μmol mmol$^{-1}$, 0.023 mol mol$^{-1}$, and 5.59 μmol g$^{-1}$ s$^{-1}$ for *WUE*, *LUE*, and *NUE* for *A. ordosica* and 1.47 μmol mmol$^{-1}$, 0.011 mol mol$^{-1}$, and 2.04 μmol g$^{-1}$ s$^{-1}$ for *L. secalinus* (Figure 5). Both species had significant seasonal differences in *WUE*, *LUE*, and *NUE* (Figure 5A–C), with CV of 43, 52, and 37% for *A. ordosica* and 53, 67, and 35% for *L. secalinus* (Figure 5).

Monthly *WUE* lowered in summer, with minima of 0.90 and 0.94 μmol mmol$^{-1}$ in July for *A. ordosica* and *L. secalinus* (Figure 6A), respectively. Nitrogen use efficiency peaked in summer for both species, with maxima of 6.97 and 2.65 μmol g$^{-1}$ s$^{-1}$ in August (Figure 6C). Light use efficiency peaked at different months of the growing season for the two species, i.e., *LUE* being maximum in spring for *A. ordosica*, with a value of 0.033 mol mol$^{-1}$, and in summer for *L. secalinus*, with a value of 0.016 mol mol$^{-1}$ (Figure 6B).

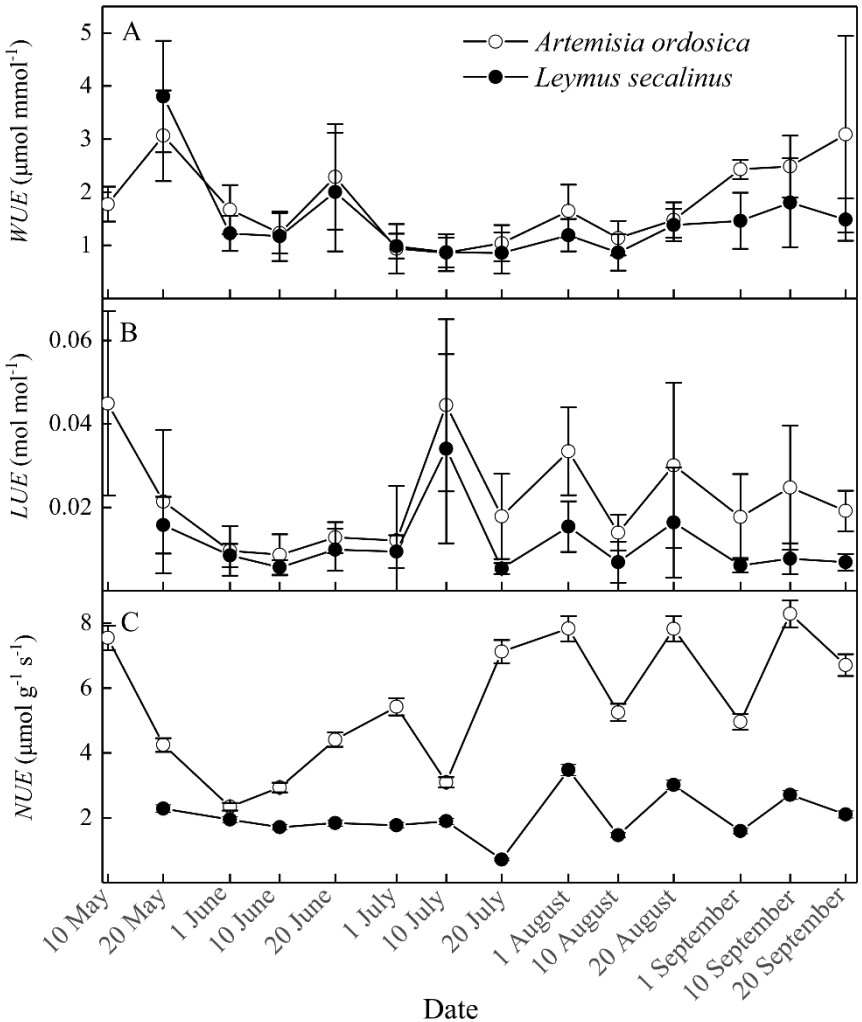

**Figure 5.** Seasonal changes in water (**A**), light (**B**), and nitrogen use efficiencies (**C**) (i.e., *WUE*, *LUE*, and *NUE*) in *A. ordosica* and *L. secalinus*. Bars indicate standard error of estimate.

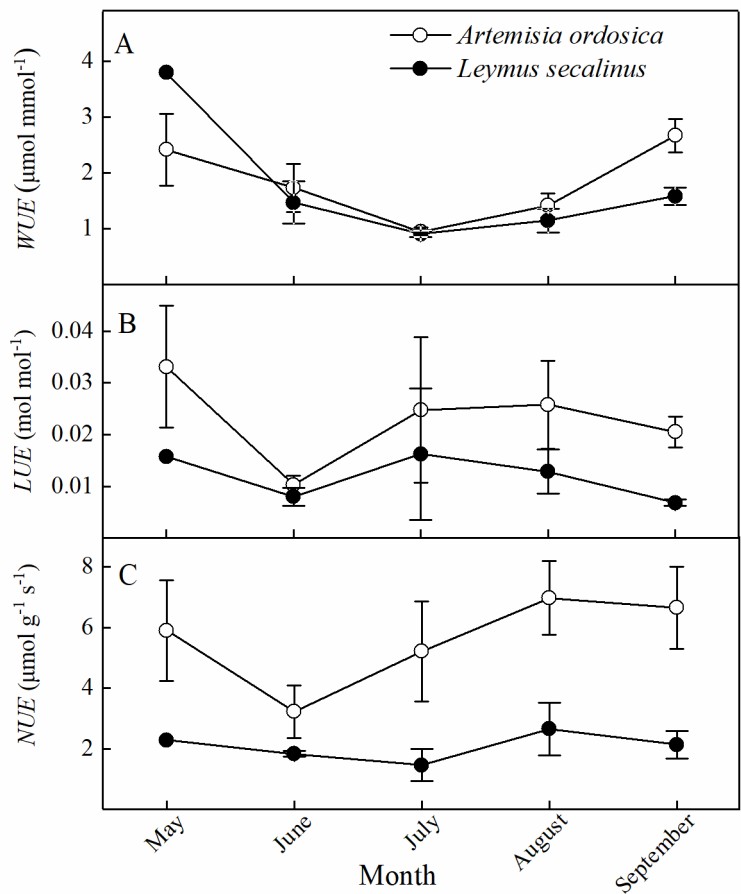

**Figure 6.** Monthly water (**A**), light (**B**), and nitrogen use efficiencies (**C**) (i.e., *WUE, LUE,* and *NUE*) from May−September 2109. Bars indicate standard error of estimate.

*3.3. Relationships among RUEs for a Given Species and between Species*

Positive correlations were detected between *LUE* and *NUE* for both *A. ordosica* ($r^2 = 0.77$, $p < 0.01$；Figure 7C) and *L. secalinus* ($r^2 = 0.94$, $p < 0.01$; Figure 7F). There were positive correlations between *WUE* and *LUE* ($r^2 = 0.65$, $p < 0.01$; Figure 7A) for *A. ordosica* and *WUE* and *NUE* for *L. secalinus* ($r^2 = 0.81$, $p < 0.01$; Figure 7E), but no such correlation existed when comparing the opposite use efficiency pairings (Figure 7B,D). *RUEs* for *A. ordosica* were positively correlated with those for *L. secalinus* (*WUE*, $r^2 = 0.40$, $p < 0.01$; *LUE*, $r^2 = 0.18$, $p < 0.01$; and *NUE*, $r^2 = 0.23$; $p < 0.01$; Figure 8). Overall, there were statistically significant differences in *WUE, LUE,* and *NUE* between the two species ($p < 0.01$, based on student t-tests; Figure 9).

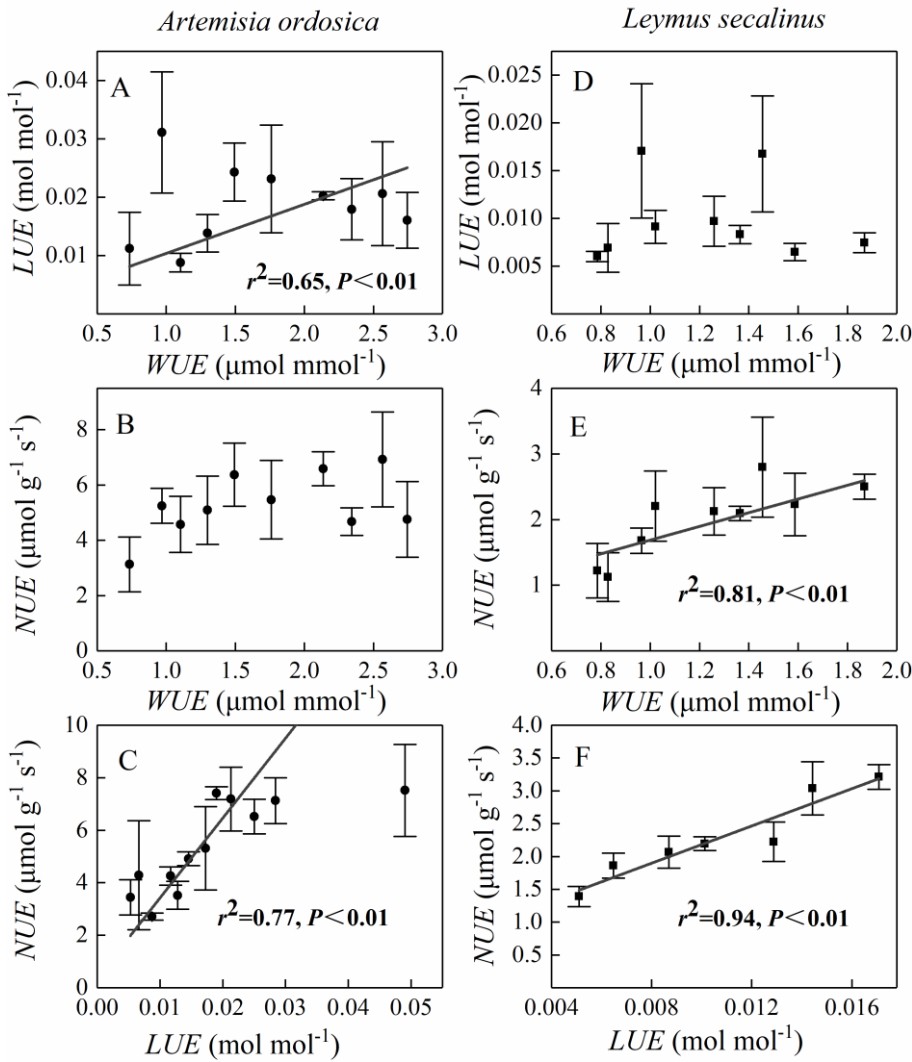

**Figure 7.** Relationships between resource use efficiencies (i.e., *WUE*, *LUE*, and *NUE*) during the growing season. Relationships between *LUE* and *WUE* (**A**), *WUE* and *NUE* (**B**), *LUE* and *NUE* (**C**) for *A. ordosica* and relationships between *LUE* and *WUE* (**D**), *WUE* and *NUE* (**E**), *LUE* and *NUE* (**F**) for *L. secalinus*. Data points are binned averages, with *WUE* and *LUE* specified in increments of 0.2 μmol mmol⁻¹ and 0.002 mol mol⁻¹, respectively. Bars indicate standard error of estimate.

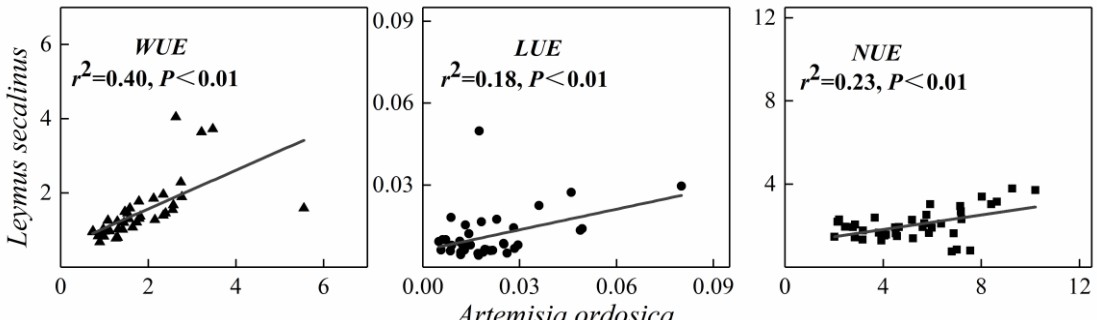

**Figure 8.** Pairwise correlations of *WUE*, *LUE*, and *NUE* for *A. ordosica* and *L. secalinus*; *R* is the coefficient of correlation and *p* provides the level of significance.

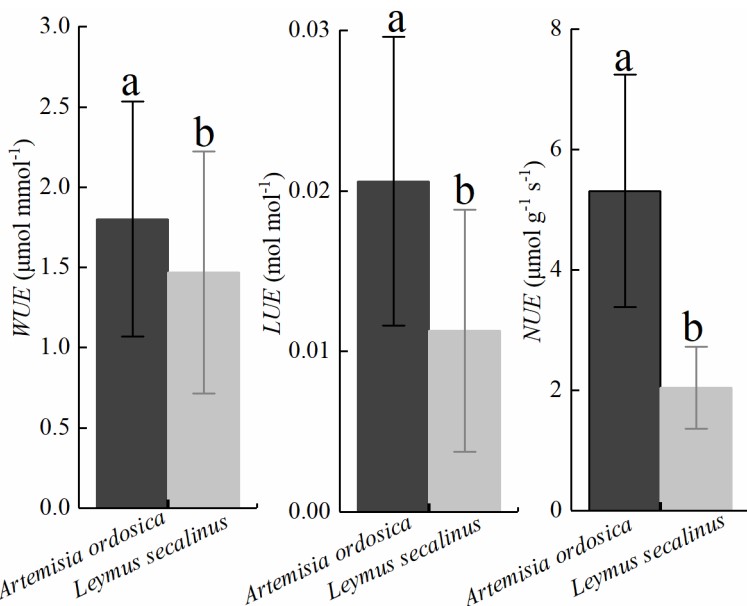

**Figure 9.** Pairwise comparisons of *WUE*, *LUE*, and *NUE* for *A. ordosica* and *L. secalinus*. Different letters (a and b) indicate statistically significant differences by paired, student t-tests based on a critical probability of 0.01.

### 3.4. Controlling Factors on Variations in RUEs

Resource use efficiencies were mainly affected by *Gs*, *SWC*, and *VPD* for both species (Table 1). Specific leaf area only affected *RUE*s in *L. secalinus*. Control of biophysical factors on *RUE*s was through their direct and indirect effects on *P*n, *E*, and $N_{area}$ (Supplementary Material Figures S1A–C and S2A–C). Moreover, *RUE*s were essentially more strongly regulated by *P*n than by *E*, *PAR*, or $N_{area}$ (Supplementary Material Table S1). Net photosynthesis was more directly affected by *Gs* and *VPD*, while *E* was more directly affected by *Gs* and indirectly affected by *SWC*. Nitrogen concentration by leaf area was more directly controlled by *SLA*. An application of SEM further demonstrated that *RUE*s were mostly affected by *VPD*, *SWC*, and *Gs*, among which *Gs* affected *RUE*s the most in both species (Supplementary Material Figures S1D–F and S2D–F).

**Table 1.** Results of stepwise regression on the relationships between *RUE*s and the biophysical factors associated with *A. ordosica* and *L. secalinus*.

|  | **RUEs** | **Model** | **$R^2$** | **F** | **p** |
|---|---|---|---|---|---|
| *Artemisia ordosica* | *WUE* | y = −1.40*VPD*–0.71*Gs* + 3.50 | 0.24 | 7.55 (1,40) | 0.002 |
|  | *LUE* | y = 0.42*SWC*+0.08*Gs* + 0.03 | 0.32 | 10.46 (1,40) | <0.001 |
|  | *NUE* | Y = 8.99*Gs* + 2.58 | 0.56 | 53.86 (1,40) | <0.001 |
| *Leymus secalinus* | *WUE* | y = −0.01*SLA*−2.98*Gs*−1.25*VPD* + 4.19 | 0.37 | 8.38 (1,37) | <0.001 |
|  | *LUE* | y = 0.24*SWC* + 0.16*Gs* + 1.74 | 0.41 | 17.66 (1,37) | <0.001 |
|  | *NUE* | y = 0.02*SLA* + 3.88*Gs* + 0.06 | 0.36 | 11.46 (1,37) | <0.001 |

$R^2$ is the coefficient of determination, *F* the *F*-ratio, and *p* the level of significance. Numbers in parentheses represent numerator and denominator degrees of freedom.

## 4. Discussion

### 4.1. Variations in RUEs and Their Controlling Factors

Variation in plant *RUE*s can indicate plant growth strategies in different environmental conditions [37,38]. Previous studies have reported that environmental fluctuations can cause changes in *RUE*s [16,17,26]. Our finding that *RUE*s is significantly affected by *SWC* and *VPD* over the growing season (Table 1), is supported by previous findings in arid-shrub species, such as *S. psammophila* and *H. mongolicum* [26,39]. Soil water content and

*VPD* determined seasonal variation in *RUE*s by controlling *P*n and *E*, but more predominantly *P*n (Supplementary Material Figures S1A–C and S2A–C). High *VPD* induced stomatal closure, preventing excessive water loss, resulting in decreased *G*s and, thus, a disproportionately larger decrease in *P*n, than in *E* [40–42]. This led to a lowering of *WUE*, as shown by its negative relationship with *VPD* and *G*s (Table 1). Soil water boosted by intermittent rain pulses, normally resulted in an elevated *E*, more than observed in *P*n, leading to a reduction in *WUE* during the summer (Supplementary Material Figures S1D and S2D) [43–45].

The result that *LUE* was positively controlled by *G*s and *SWC* (Table 1), is commonly observed in desert plants [46–48], for which light is not a limiting factor. Soil water supply in dry areas is often improved by intermittent rain pulses, leading to larger photosynthetic capacities [36]. When soil water was enough to meet atmospheric demand, *P*n increased along with *G*s (Supplementary Material Figures S1A–C and S2A–C) [9,45], causing *LUE* to be greater, and vice versa. Offset of water restrictions increased the efficiency by which *PAR* was converted to photosynthates. A reduction in stomatal constraint probably also played a positive role.

Since $N_{area}$ changed very little over the growing season, the observed seasonality in *NUE* was mainly due to changes in *P*n (Figures 3A and 4C; Supplementary Material Table S1), establishing a positive relationship between *NUE* and *G*s through its positive effect on *P*n (Table 1). Higher stomatal conductance had caused *P*n to be higher [11], which may have caused *NUE* to be greater during the summer.

Overall, variations in *RUE*s were largely controlled by *G*s. Stomatal conductance was itself mostly controlled by *SWC* and *VPD*. Seasonal patterns in *RUE*s were due to the control applied by intermittent rain pulses and associated soil water supplies. This result confirmed that the presence of water was responsible for the large changes observed in resource use and efficiencies in arid and semiarid grasslands [10,13,49,50]. It submits that drier conditions at our site more likely decreased *SWC* in situ, which limited plant physiological activity and growth. It may be important for vegetation-climate models of net primary productivity to address these fundamentals in predicting plant responses in dryland ecosystems.

*A. ordosica* and *L. secalinus*, as indicator species, can be selected on the basis of their trait values' responsiveness to environmental factors and their importance both locally and regionally [9,19,39], for monitoring trends in ecosystem-level properties across environmental gradients (e.g., pollution, drought, fertility) [51]. These field measurements always involve a balance between the number of replicates and precision. The number of replicates selected should depend on species variability in the trait of interest, as well as on the number of species sampled [34]. Pérez-Harguindeguy et al. showed that the minimum and preferred number of replicates for different traits is five and ten, respectively, based on common practice [34]. Prior studies in semiarid shrublands usually involved three to seven replicates [26,35,39,45]. Given constraints of time, we selected nine replicates for each species for an improved assessment, and thus could guarantee validity of our study results.

### 4.2. Tradeoffs between RUEs

High *NUE* occurred predominantly when *LUE* was high in both species (Figure 7C,F), such as those seen in boreal trees [52], suggesting that a level of convergence existed between *LUE* and *NUE*. It was previously reported that plants can reduce the constraints on carbon uptake by maximizing the use efficiency of the most limiting resource, while lowering the use efficiency of resources that are more abundant [53]. Some prior studies have shown a negative association between *LUE* and *NUE*, in contrast to our results [16,54]. One plausible explanation for this discrepancy may be associated with the fact that *LUE* in plants reflect processes of carbon fixation. This is not the case with *NUE*, where *NUE* is mostly the product of carbon fixation and protein synthesis, which are

weakly coupled to carbon uptake. Temporal integration also tends to decrease the relative importance of resource availabilities and cause long-term *RUE* responses to differ from short-term observations [52,55]. As a key ecological function, long-term relative changes in *RUE* deserve further investigation for improved understanding in how plants may respond to climate change, especially with long-lasting effects of highly variable precipitation and extreme aridity in dryland ecosystems.

In *A. ordosica*, variation in *NUE* and *LUE* were mostly explained by variation in *P*n, leading to a positive correlation between *LUE* and *NUE* (Supplementary Material Table S1). Increased *P*n was accompanied by an elevated *G*s, increasing *LUE* and *NUE* when leaf *N* concentrations were relatively stable [56]. In *L. secalinus*, positive correlation between *LUE* and *NUE* were explained by variations in *P*n and $N_{mass}$ (Figures 3A and 4B; Supplementary Material Table S1). The plasticity of *LUE* to high insolation levels was largely due to *P*n being limited by carboxylation capacity and associated $N_{mass}$ [57]. High insolation acted to maximize instantaneous photosynthesis, which correlated positively with *LUE* due to reduced absorption of saturated light, and positively with *NUE* due to lower *N* investment that maximized *P*n during the summer [58,59].

There was neither correlation between *WUE* and *LUE*, nor correlation between *WUE* and *NUE* in both species (Figure 7A,B,D,E). The Mu Us Desert is occasionally affected by drought; *SWC* was shown to be a key factor in controlling plant *RUE*s and constraining plant growth (Table 1) [10,26,45,50]. *A. ordosica* has an ability to avoid functional damage by reducing transpiration losses through stomatal closure during periods of excessive dryness [9,39]. Consequently, its *RUE*s response to drought largely depended on the physiological control conveyed by *G*s. Physiological and structural variations in *G*s and *SLA* both regulated *RUE*s in *L. secalinus* (Table 1). Drought-coping variations in plant morphological and physiological traits could have led to differences in tradeoffs between *RUE*s in the two species, as observed in prior studies [18,29,60]. The results indicated that dominant species have adaptive differentiation of resource use-related traits to achieve local coexistence in dryland ecosystems.

### 4.3. Relationships in RUEs between the Two Species

The finding that *RUE*s in *A. ordosica* was positively and linearly correlated with those in *L. secalinus* (Figure 8), provides some evidence of convergence. This result is consistent with trends seen in other studies [e.g., 26,29,38], such as those reported for piñon pine and juniper [11]. *RUE*s in dominant plant species growing in harsh environments, such as arid [61], semiarid Mediterranean [38], and semiarid shrubland (this study) are subject to the effects of various levels of drought intensity and duration. Clearly, convergence in *RUE*s in the dominant plant species can be largely explained by soil water limitations experienced at these sites. It further confirms that ecosystem functioning is more sensitive and vulnerable to highly variable precipitation, extreme water scarcity, and pronounced fluctuations in diurnal temperatures in drylands [2,4].

Although convergence existed in *RUE*s between the two species, there was a difference in their magnitudes. *RUE*s in *A. ordosica* were mostly greater than those in *L. secalinus* (Figures 5 and 6; $p < 0.01$, Figure 9), due to their elevated leaf photosynthetic capacity and *P*n (Figure 3A). Prominent *SLA* in *A. ordosica* means that it has a larger capacity to capture light and acquire nutrients more directly than in *L. secalinus*. These results suggest that shrubs may be better suited for arid conditions, as previously proposed by Zha et al. and Wu et al. [9,26,36]. Compared to *RUE*s in *L. secalinus*, variation in *RUE*s was more limited in *A. ordosica* (Figure 5A,B), indicating that desert shrubs may be more resistant to drought. Suppression of *SLA*, $N_{mass}$, and $N_{area}$ was greatest in *A. ordosica*, compared to that in *L. secalinus* (Figure 4). Soil water content (*SWC*) and *VPD* had more influence on *LUE* and *WUE* in *L. secalinus*, suggesting that the species is more responsive to drought than *A. ordosica*. Past studies have reported similar findings in other desert plants (e.g., [9,19,62]).

There was divergence in the relationship between plant drought tolerance and its *RUE*s [11]. Desert plants either select resource acquisition or resource conservation in their

response to drought [63]. In this study, *A. ordosica* demonstrated a resource-conservation strategy [10,26,39]. This explains the high *RUE*s and low sensitivity to environmental change observed in *A. ordosica*, compared with *L. secalinus* (Figures 5 and 6; Supplementary Material Figures S1D–F and S2D–F). *A. ordosica* had higher *RUE*s, and thus could accelerate phenological change, improve photosynthetic production, and complete its life cycle as quickly as possible to avoid dehydration and escape drought, which resulted in its better acclimatization to dry environments in comparison with *L. secalinus* [64]. With continued increases in drought severity and duration and associated lowering of groundwater reserves anticipated with future climate change, desert shrubs are projected to eventually replace grasses, impairing the function of natural ecosystems and, thus, diminishing the number and quality of ecological services they provide [2,65]. Our results suggest that the preservation of plant biodiversity in arid and semiarid drylands is crucial to buffer the undesirable effects of climate change and desertification.

## 5. Conclusions

Both species had similar seasonal patterns in *RUE*s (i.e., *WUE*, *LUE*, and *NUE*) and exhibited significant seasonal variation in *RUE*s from May–September 2019 (CV > 30%). Monthly *WUE* lowered in July and *NUE* peaked in August for both *A. ordosica* and *L. secalinus*. Light use efficiency peaked at different months for the two species. *SWC* and *VPD* were the environmental factors that affected variation in *RUE*s the most by regulating *G*s. For a given *RUE*, there was a convergence in resource use efficiency between the two species, with *A. ordosica* exhibiting lower overall variation compared to *L. secalinus*. However, the tradeoffs between *RUE*s were divergent in the species. It was determined that *A. ordosica* was more adaptable to arid conditions than *L. secalinus*. Vegetation succession may lead to shifts in ecosystem composition in favor of more drought tolerant species, such as *A. ordosica*, in the near-to-immediate future. This may result in a reduction in biodiversity and ecosystem functioning, and enhanced vulnerability to global climate change and human disturbance in drylands. Dryland social-ecological systems are especially sensitive to rapid rates of physical and social change, such as those associated with climate change and urbanization. Our findings demonstrate the need to strengthen dryland ecosystem management methods for sustainable livelihoods and to advance progress towards the implementation of 2030 SDGs for drylands.

**Supplementary Materials:** The following are available online at www.mdpi.com/article/10.3390/f12101372/s1. Figure S1: Structural equation modelling (SEM, subfigures A–C) and standardized total effect (D–F) showing the effect of abiotic and biotic factors on *RUE*s in *A. ordosica*; Figure S2: Structural equation modelling (SEM, subfigures A–C) and standardized total effect (D–F) showing the effect of abiotic and biotic factors on *RUE*s in *L. secalinus*; Table S1: Partial correlation coefficients between *RUE*s [i.e., water, light, and nitrogen use efficiencies (i.e., *WUE*, *LUE*, and *NUE*)] and biophysical variables of photosynthesis (*P*n), transpiration (*E*), incident photosynthetically active radiation (*PAR*), and N per unit leaf area (*N*$_{area}$) from 1 May–30 September 2019 for *A. ordosica* and *L. secalinus*.

**Author Contributions:** Y.T. and Y.J. conceived the study. Y.J., C.J., N.W., and S.G. conducted the fieldwork. Y.J., X.J. (Xiaoyan Jiang) and X.L. analyzed the data. Y.J. wrote the manuscript with the assistance of Y.T. and T.Z. T.Z., X.J. (Xin Jia), and P.L. revised the manuscript. C.P.-A.B. revised the initial and final writing of the manuscript. All authors have read and agreed to the published version of the manuscript.

**Funding:** This work was supported by the National Natural Science Foundation of China (NSFC: 31901366, 32071842, 32071843), the National Key Research and Development Program of China (2020YFA0608100), and by the Fundamental Research Funds for the Central Universities (No. 2015ZCQ-SB-02).

**Institutional Review Board Statement:** Not applicable.

**Informed Consent Statement:** Not applicable.

**Data Availability Statement:** The data and materials that support the findings of this study are all available from the corresponding author upon reasonable request.

**Acknowledgments:** We would like to thank X.W. Yang, S.J. Liu, G.P. Chen, C. Zhang, Y. Luo and H. Tian for their assistance with field measurements and instrument maintenance. The U.S.-China Carbon Consortium (USCCC) supported this work by way of helpful discussion and the exchange of scientific ideas.

**Conflicts of Interest:** The authors declare no conflicts of interest.

## Appendix A

**Table A1.** Acronyms addressed in this study and their units of measurement.

| Full-Name | Abbreviation | Unit |
|---|---|---|
| Resource use efficiencies | *RUE*s | - |
| Water use efficiency | *WUE* | $\mu mol \cdot mmol^{-1}$ |
| Light use efficiency | *LUE* | $mol \cdot mol^{-1}$ |
| Nitrogen use efficiency | *NUE* | $\mu mol \cdot g^{-1} \cdot s^{-1}$ |
| Photosynthetic rate | $Pn$ | $\mu mol \cdot m^{-2} \cdot s^{-1}$ |
| Transpiration rate | $E$ | $mmol \cdot m^{-2} \cdot s^{-1}$ |
| Stomatal conductance | $Gs$ | $mol \cdot m^{-2} \cdot s^{-1}$ |
| Specific leaf area | *SLA* | $cm^2 \cdot g^{-1}$ |
| Leaf nitrogen content per unit area | $N_{area}$ | $g \cdot m^{-2}$ |
| Leaf nitrogen per unit dry mass | $N_{mass}$ | $g \cdot kg^{-1}$ |
| Air temperature | $T$ | $^\circ C$ |
| Relative humidity | *RH* | % |
| Net radiation | Rn | $W \cdot m^{-2}$ |
| Incident photosynthetically active radiation | *PAR* | $\mu mol \cdot m^{-2} \cdot s^{-1}$ |
| Soil water content at 30-cm depth | SWC | $m^3 \cdot m^{-3}$ |
| Vapor pressure deficient | *VPD* | kPa |
| Coefficient of variation | CV | % |

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
