# Peer review of "Dynamic Changes in Plant Resource Use Efficiencies and Their Primary Influence Mechanisms in a Typical Desert Shrub Community"

_forests, doi:10.3390/f12101372_

Round 1

Reviewer 1 Report

This is an ambitious and significant study that nevertheless has flaws. By relying on instantaneous measurements between 0800 and 1800 to calculate WUE and LUE, the authors miss an important part of the day when light intensity and temperatures are lower. At these times WUE and LUE is likely to be higher. So the monthly averages of WUE and LUE that are derived from the instantaneous measures are likely to be biased as well. It would be helpful if the efficiency measures were calculated using integrated quantities such the daily light integral (DLI) and daily carbon gain, which is photosynthesis minus respiration. A good measure of integrated WUE is provided by ratio of stable 13C to 12C. The lack of such measures does not disqualify the present study, but I encourage the authors to consider them in the future.

I am surprised to see such variation in SLA, particularly for A. ordosica. is this because leaves are produced continuously through the season?

It is not surprising to me that RUE’s converge since both species are C3 plants in arid landscape and the RUE’s are largely determined by stomatal behavior and the effect of VPD on transpiration. I am not convinced that greater RUE confers an adaptive advantage in a competitive setting. If the species with lower RUE uses most of the resource first then there is none left for the more efficient species. This reasoning seems to apply to WUE particularly. In short, greater RUE may only be advantageous in monospecific stands, which are rare outside of cultivation.

Author Response

Dear Reviewer,

Thank you very much for the careful review of our manuscript (Forests-1382254), entitled “Dynamic changes in plant resource use efficiencies and their primary influence mechanisms in a typical desert shrub community.” The reviewer’s comments are very much appreciated. The manuscript has been carefully revised in light of all the comments. Point-by-point responses to each comment are addressed below.

All changes made in the revised manuscript are shown in track-change mode, such that any changes can be easily viewed by you in the changes-marked version of the revised manuscript. Please do not hesitate to contact me if questions arise.

Yours sincerely,

Dr. Yun Tian

Note: all line numbers mentioned below represent those in the clean version of the revised manuscript.

*This is an ambitious and significant study that nevertheless has flaws. By relying on instantaneous measurements between 0800 and 1800 to calculate WUE and LUE, the authors miss an important part of the day when light intensity and temperatures are lower. At these times WUE and LUE is likely to be higher. So the monthly averages of WUE and LUE that are derived from the instantaneous measures are likely to be biased as well. It would be helpful if the efficiency measures were calculated using integrated quantities such the daily light integral (DLI) and daily carbon gain, which is photosynthesis minus respiration. A good measure of integrated WUE is provided by ratio of stable 13C to 12C. The lack of such measures does not disqualify the present study, but I encourage the authors to consider them in the future.

Reply:

We agree with the reviewer. We would consider your advice in a future study.

*I am surprised to see such variation in SLA, particularly for A. ordosica. is this because leaves are produced continuously through the season?

Reply:

SLA reflects the ability of plants to acquire resources and mainly expresses the balance between carbon acquisition and utilization of plants (Chen et al., 2016; Wang et al., 2017). Because of that the specific leaf area (SLA) was calculated as the ratio of fresh leaf area to dry weight, changes of leaf shape and size affected the SLA. With the expansion of the leaves in summer, high values of SLA exhibited as a result of their shape and size being thinner and bigger, respectively, which improved the capability to capture light and acquire nutrients to promote vigorous growth (Chen et al., 2015). It has been previously reported that SLA is positively correlated with precipitation (Gouveia and Freitas, 2009). Adaptive variation of SLA was determined by different resource availability gradients (Brouillette et al., 2014). So SLA fluctuated irregularly without always keeping high values in summer and exhibiting high values in September, maybe as a result of frequent drought. Drought persisted in September (Figure 2D). As leaf-coloring occurred in September, A. ordosica had high SLA, which led to an accelerated uptake of light and nutrients, in a push to complete its life cycle early to escape drought in September.

  1. Cheng, J., Chu, P., Chen, D., and Bai, Y. Functional correlations between specific leaf area and specific root length along a regional environmental gradient in Inner Mongolia grasslands. Ecol. 2016, 30, 985-997. doi: 10.1111/1365-2435.12569
  2. Wang, C.Y.; Xiao, H.G.; Liu, J.; Zhou, J.W. Differences in leaf functional traits between red and green leaves of two evergreen shrubs Photinia × fraseri and Osmanthus fragrans. Forestry Res. 2017, 28, 473-479. doi:10.1007/s11676-016-0346-7
  3. Chen, Z.-H.; Zha, T.; Jia, X.; Wu, Y.; Wu, B.; Zhang, Y.; Guo, J.; Qin, S.; Chen, G.; Peltola, H. Leaf nitrogen is closely coupled to phenophases in a desert shrub ecosystem in China. Arid Environ. 2015, 122, 124-131. doi: 10.1016/j.jaridenv.2015.07.002
  4. Gouveia, A.C.; Freitas, H. Modulation of leaf attributes and water use efficiency in Quercus suber along a rainfall gradient. Trees 2009, 23, 267-275. doi:10.1007/s00468-008-0274-z
  5. Brouillette, L.C.; Mason, C.M.; Shirk, R.Y.; Donovan, L.A. Adaptive differentiation of traits related to resource use in a desert annual along a resource gradient. Phytol. 2014, 201, 1316-1327. doi: 10.1111/nph.12628

*It is not surprising to me that RUE’s converge since both species are C3 plants in arid landscape and the RUE’s are largely determined by stomatal behavior and the effect of VPD on transpiration. I am not convinced that greater RUE confers an adaptive advantage in a competitive setting. If the species with lower RUE uses most of the resource first then there is none left for the more efficient species. This reasoning seems to apply to WUE particularly. In short, greater RUE may only be advantageous in monospecific stands, which are rare outside of cultivation.

Reply:

We agree with the reviewer that greater RUE may only be advantageous in monospecific stands, which are rare outside of cultivation. But there was divergence in the relationship between plant drought tolerance and its resource use efficiency (Limousin et al., 2015). Desert plants either select resource acquisition or resource conservation in their response to drought (Brouillette et al., 2014). In this study, A. ordosica demonstrated a resource-conservation strategy (Chen et al., 2015; Wu et al., 2018; Iqbal et al., 2021). This explains the high RUEs and low sensitivity to environmental change observed in A. ordosica, compared with L. secalinus (Figures 5, 6; Supplementary Material Figures S1, S2). A. ordosica had higher RUEs, and thus could accelerate phenological change, improve photosynthetic production, and complete its life cycle as quickly as possible to avoid dehydration and escape drought (Ivey & Carr, 2012). Because of its rapid growth, A. ordosica can avoid the competition with L. secalinus in the resources, no matter how many resources that L. secalinus used, especially during long dry spells. And lower RUEs of L. secalinus mean that it needs more resources to normal growth, leading to no enough resources to maintain physiological activities, and thus competitive power declined compared to A. ordosica under drought stress. Consequently, A. ordosica is anticipated to take a resource-conservative strategy in its better acclimatization to dry environments. We clarified this in the revised manuscript (lines 416-424 in clean version of revised manuscript).

  1. Limousin, J.M.; Yepez, E.A.; McDowell, N.G.; Pockman, W.T.; Tjoelker, M. Convergence in resource use efficiency across trees with differing hydraulic strategies in response to ecosystem precipitation manipulation. Funct. Ecol. 2015, 29, 1125-1136. doi: 10.1111/1365-2435.12426
  2. Brouillette, L.C.; Mason, C.M.; Shirk, R.Y.; Donovan, L.A. Adaptive differentiation of traits related to resource use in a desert annual along a resource gradient. New. Phytol. 2014, 201, 1316-1327. doi: 10.1111/nph.12628
  3. Chen, Z.-H.; Zha, T.; Jia, X.; Wu, Y.; Wu, B.; Zhang, Y.; Guo, J.; Qin, S.; Chen, G.; Peltola, H. Leaf nitrogen is closely coupled to phenophases in a desert shrub ecosystem in China. J. Arid Environ. 2015, 122, 124-131. doi: 10.1016/j.jaridenv.2015.07.002
  4. Wu, Y.J.; Ren, C.; Tian, Y.; Zha, T.S.; Liu, P.; Bai, Y.J.; Ma, J.Y.; Lai, Z.R.; Bourque, C.P.A. Photosynthetic gas-exchange and PSII photochemical acclimation to drought in a native and non-native xerophytic species (Artemisia ordosica and Salix psammophila). Ecol. Indic. 2018, 94, 130-138. doi: 10.1016/j.ecolind.2018.06.040
  5. Iqbal, S.; Zha, T.; Jia, X.; Hayat, M.; Qian, D.; Bourque, C.P.A.; Tian, Y.; Bai, Y.; Liu, P.; Yang, R.; et al. Interannual variation in sap flow response in three xeric shrub species to periodic drought. Agric. Forest Meteorol. 2021, 297, 108276. doi: 10.1016/j.agrformet.2020.108276
  6. Ivey CT, Carr DE. 2012. Tests for the joint evolution of mating system and drought escape in Mimulus. Annals of Botany 109: 583-598

Reviewer 2 Report

Dear Authors

Please find comments in the attached file.

Regards

Author Response

Dear Reviewer,

Thank you very much for the careful review of our manuscript (Forests-1382254), entitled “Dynamic changes in plant resource use efficiencies and their primary influence mechanisms in a typical desert shrub community.” The reviewer’s comments are very much appreciated. The manuscript has been carefully revised in light of all the comments. Point-by-point responses to each comment are addressed below.

All changes made in the revised manuscript are shown in track-change mode, such that any changes can be easily viewed by you in the changes-marked version of the revised manuscript. Please do not hesitate to contact me if questions arise.

Yours sincerely,

Dr. Yun Tian

Note: all line numbers mentioned below represent those in the clean version of the revised manuscript.

*Introduction. The problem of your research has to be formulated more clearly.

Reply:

We’ve revised our Introduction as recommended by the reviewer.

(Introduction in clean version of revised manuscript).

*Methods.

(1) In introduction part you have formulated three tasks of your work. In this part you have to clarify not only the fields, measurements and other, but also clearly indicate the used methods for formulated tasks. This would increase understanding of your paper and will contribute to its structure.

Reply:

We clearly indicate the methods used in formulating the tasks in the revised manuscript as follows (see lines 194 - 204 in clean version of revised manuscript).

The coefficient of variation (CV) was used to quantify the seasonal variation in RUEs (i.e., WUE, LUE, and NUE) and biotic factors. To analyze relationships among WUE, LUE, and NUE for a given species and to determine whether a level of convergence exists in RUEs between the two plant species, a standard major axis (SMA) operation was performed using sma, a procedure available in the smatr R-package. Paired student t-tests were performed in pairwise comparisons of WUE, LUE, and NUE for A. ordosica and L. secalinus. To clarify the role of factors in the control of seasonal dynamics in RUEs, stepwise regression was used to find the most critical biophysical factors responsible for controlling RUEs. In addition, structural equation models (SEM) were used to assess the direct and indirect contributions of biotic and abiotic factors to variations in RUEs. The significance level was set at P= 0.05.

(2) No site replications of your study

Reply:

There are no site replicates of our study. But our study site is representative of a transitional zone between arid and semiarid conditions at the southern edge of the Mu Us Desert. The landscape of this region is typical inland dune ecosystems with very distinct habitat types (She et al., 2017). And 100 m×100 m plot was established at this site. We chose three 5×5 m2 subplots from the established plot, which can greatly reflected Artemisia ordosica -dominated plant communities. So the study in this site was representative.

She, W.; Bai, Y.; Zhang, Y.; Qin, S.; Liu, Z.; Wu, B. Plasticity in meristem allocation as an adaptive strategy of a desert shrub under contrasting environments. Front. Plant Sci. 2017, 8, 1933. doi: 10.3389/fpls.2017.01933

(3) You have to convince the reviewer and the other readers that 9 measurements per species is enough to answer formulated tasks. It is very important for acknowledging plausibility of your results. Here you could provide references supporting your used method.

Reply:

We now address the feasibility of the 9 measurements per species (see lines 342 - 353 in clean version of revised manuscript) and provide references supporting our methods used in the revised manuscript (lines 146 -148 in clean version of revised manuscript).

*Result. You present many results, but it is difficult to understand which results belongs to which task. Please relate your results to formulated research tasks by structuring your results part according to formulated research tasks.

Reply:

We have reconstructed our Results section of the manuscript according to formulated research tasks. Section 3.1 describes the variations in biophysical factors and photosynthetic parameters, which provide some statistical basis for our analysis (lines 211-243 in clean version of revised manuscript). Section 3.2 addresses Task 1 w.r.t. examining seasonal dynamics in the individual components of RUEs (lines 244 -264 in clean version of revised manuscript). Section 3.3 addresses Task 2 w.r.t. determining whether a level of convergence exists in RUEs between plant species (lines 265 -288 in clean version of revised manuscript), and section 3.4 addresses Task 3 w.r.t. clarifying the role of environmental factors in the control of seasonal dynamics in RUEs (lines 289 -304 in clean version of revised manuscript).

*Discussion. please show the importance of your findings in relation to other research. Also discuss the validity of your results since you used only 9 trees per species.

Reply:

We show the importance of our findings in relation to other research in light of your advice (lines 337 -341, 366-369, 389-391, 400-402, & 428-430 in clean version of revised manuscript). We now discuss the validity of our results with 9 replicates per plant species (lines 342-353 in clean version of revised manuscript) as follows.

  1. ordosica and L. secalinus, as indicator species, can be selected on the basis of their trait values’ responsiveness to environmental factors and their importance both locally and regionally (Zha et al., 2017; Jia et al., 2018; Iqbal et al., 2021), for monitoring trends in ecosystem-level properties across environmental gradients (e.g., pollution, drought, fertility) (De Bello et al., 2011). These field measurements always involve a balance between the number of replicates and precision. The number of replicates selected should depend on species variability in the trait of interest, as well as on the number of species sampled (Pérez-Harguindeguy et al., 2016). Pérez-Harguindeguy et al. showed that the minimum and preferred number of replicates for different traits is 5 and 10, respectively, based on common practice (Pérez-Harguindeguy et al., 2016). Prior studies in semiarid shrublands usually involved 3 to 7 replicates (Ren et al., 2018; Wu et al., 2018; Hayat et al., 2020; Iqbal et al., 2021). Given constraints of time, we selected nine replicates for each species for an improved assessment, and thus could guarantee validity of our study results.

  1. Zha, T.; Qian, D.; Jia, X.; Bai, Y.; Tian, Y.; Bourque, C.P.A.; Ma, J.; Feng, W.; Wu, B.; Peltola, H. Soil moisture control of sap-flow response to biophysical factors in a desert-shrub species, Artemisia ordosica. Biogeosciences 2017, 14, 4533-4544. doi: 10.5194/bg-14-4533-2017
  2. Jia, X.; Zha, T.; Gong, J.; Zhang, Y.; Wu, B.; Qin, S.; Peltola, H. Multi-scale dynamics and environmental controls on net ecosystem CO2 exchange over a temperate semiarid shrubland. Forest Meteorol. 2018, 259, 250-259. doi: 10.1016/j.agrformet.2018.05.009
  3. Iqbal, S.; Zha, T.; Jia, X.; Hayat, M.; Qian, D.; Bourque, C.P.A.; Tian, Y.; Bai, Y.; Liu, P.; Yang, R.; et al. Interannual variation in sap flow response in three xeric shrub species to periodic drought. Forest Meteorol. 2021, 297, 108276. doi: 10.1016/j.agrformet.2020.108276
  4. De Bello, F.; Lavorel, S.; Albert, C.H.; Thuiller, W.; Grigulis, K.; Dolezal, J.; Janeček, Š; Lepš, J. Quantifying the relevance of intraspecific trait variability for functional diversity. Methods in Ecology and Evolution 2011, 2, 163-174. doi:10.1111/j.2041-210X.2010.00071.x
  5. Pérez-Harguindeguy, N.; Díaz, S.; Garnier, E.; Lavorel, S.; Poorter, H.; Jaureguiberry, P.; Bret-Harte, M.S.; Cornwell, W.K.; Craine, J.M.; Gurvich, D.E.; et al. Corrigendum to: New handbook for standardised measurement of plant functional traits worldwide. Australian Journal of Botany 2016, 64, 715-716. doi:10.1071/BT12225_CO
  6. Ren, C.; Wu, Y.J.; Zha, T.S.; Jia, X.; Tian, Y.; Bai, Y.J.; Bourque, C.P.-A.; Ma, J.Y.; Feng, W. Seasonal changes in photosynthetic energy utilization in a desert shrub (Artemisia ordosica) during its different phenophases. Forests 2018, 9, 176. doi:10.3390/f9040176
  7. Wu, Y.J.; Ren, C.; Tian, Y.; Zha, T.S.; Liu, P.; Bai, Y.J.; Ma, J.Y.; Lai, Z.R.; Bourque, C.P.A. Photosynthetic gas-exchange and PSII photochemical acclimation to drought in a native and non-native xerophytic species (Artemisia ordosica and Salix psammophila). Indic. 2018, 94, 130-138. doi: 10.1016/j.ecolind.2018.06.040
  8. Hayat, M.; Zha, T.; Jia, X.; Iqbal, S.; Qian, D.; Bourque, C.P.A.; Khan, A.; Tian, Y.; Bai, Y.; Liu, P.; et al. A multiple-temporal scale analysis of biophysical control of sap flow in Salix psammophila growing in a semiarid shrubland ecosystem of northwest China. Forest Meteorol. 2020, 288-289, 107985. doi: 10.1016/j.agrformet.2020.107985

*Conclusion.

(1) Conclusions has to answer to your aim of the research and the tasks formulated. Neither of them I can find here. Please also clarify the novelty and the importance of your findings.

Reply:

We clarified our aim of the research and the tasks formulated in the Conclusions section of the revised manuscript (lines 432-436 in clean version of revised manuscript) according to your comments. We also clarify the novelty and importance of the findings (lines 442-448 in clean version of revised manuscript).

(2) This sentence has no value to your conclusions. I would delete it.

Reply:

Agreed, we have this sentence, which is the “Our findings have important implications for explaining climate-induced drought effects on RUEs and for understanding the adaptation and acclimatization capacity of dryland shrub and grass species in plant communities under current climate change.”

Reviewer 3 Report

Dear Authors, 

I thank you for the opportunity to review your article on desert shrub behavior and resource consumption. I believe it is an essential topic and one that should be covered in Forest magazine. I agree with the opinions of the authors and believe that the change in the Earth's flora and fauna is an almost irreversible fact due to such severe changes in climatology and the advance of dryness. 

However, I believe that there are several aspects of your work that can still be sensibly improved. In detail, it would be interesting to improve the following aspects: 

Introduction. It is a very short introduction and not very indicative of the state of the question. I believe that it is necessary to be somewhat more rigorous in the approaches and to expand the bibliographical references on the subject. I think it would be necessary to expand the studies that have been done on drought resistant varieties and that are cited worldwide. Therefore, I suggest that the following bibliographical references be analyzed: 

  • Fischer RA Maurer R (1978) Drought resistance in spring wheat cultivars. I. Grain yield responses . Australian Journal of Agricultural Research 29, 897-912. https://doi.org/10.1071/AR9780897 (more than 1100 citations).
  • Blum, A. (2005). Drought resistance, water-use efficiency, and yield potential - are they compatible, dissonant, or mutually exclusive? Australian Journal of Agricultural Research, 56(11), 1159-1168. doi:10.1071/AR05069 (794 citations in Scopus database)
  • Maestre, F. T., Quero, J. L., Gotelli, N. J., Escudero, A., Ochoa, V., Delgado-Baquerizo, M., . . . Zaady, E. (2012). Plant species richness and ecosystem multifunctionality in global drylands. Science, 335(6065), 214-218. doi:10.1126/science.1215442 (642 citations in Scopus database).

Also, how is your article in the framework of the Sustainable Development Goals (SDGs), and has there been any impulse to the issue you raise in your article from goals 13 and 15, for example? I am enclosing some of the most cited works worldwide for this purpose:

  • Fu, B., Stafford-Smith, M., Wang, Y., Wu, B., Yu, X., Lv, N., . . . Chen, X. (2021). The global-DEP conceptual framework — research on dryland ecosystems to promote sustainability. Current Opinion in Environmental Sustainability, 48, 17-28.  doi:10.1016/j.cosust.2020.08.009
  • Leach, M., Rockström, J., Raskin, P., Scoones, I., Stirling, A. C., Smith, A., . . . Olsson, P. (2012). Transforming innovation for sustainability. Ecology and Society, 17(2) doi:10.5751/ES-04933-170211
  • Stringer, L. C., Reed, M. S., Fleskens, L., Thomas, R. J., Le, Q. B., & Lala-Pritchard, T. (2017). A new dryland development paradigm grounded in empirical analysis of dryland systems science. Land Degradation and Development, 28(7), 1952-1961. doi:10.1002/ldr.2716

Methodology section. It would be desirable to illustrate the area in which the study was developed. The authors could consider including a map showing the territory where the study was developed. As a reader of your work, I think this would be a good idea. 

Finally, the article seems to me to be very complete and rigorous, but I think that the authors could exploit the conclusions a little more and not be so brief in this section of the article. I think it would be necessary to present a reflection on the need for governments to take note of this type of article, since it is the public administration that has to oversee this change. In this sense, it is curious to note that there is no reflection on the SDGs and the impetus given to the sustainability of the planet and the need to stop climate change. 

I wish you good luck with your article and I encourage you with the corrections. 

Author Response

Dear Reviewer,

Thank you very much for the careful review of our manuscript (Forests-1382254), entitled “Dynamic changes in plant resource use efficiencies and their primary influence mechanisms in a typical desert shrub community.” The reviewer’s comments are very much appreciated. The manuscript has been carefully revised in light of all the comments. Point-by-point responses to each comment are addressed below.

All changes made in the revised manuscript are shown in track-change mode, such that any changes can be easily viewed by you in the changes-marked version of the revised manuscript. Please do not hesitate to contact me if questions arise.

Yours sincerely,

Dr. Yun Tian

Note: all line numbers mentioned below represent those in the clean version of the revised manuscript.

* I thank you for the opportunity to review your article on desert shrub behavior and resource consumption. I believe it is an essential topic and one that should be covered in Forest magazine. I agree with the opinions of the authors and believe that the change in the Earth's flora and fauna is an almost irreversible fact due to such severe changes in climatology and the advance of dryness.

Reply:

Thank you very much, for your overall positive evaluation and encouragement.

However, I believe that there are several aspects of your work that can still be sensibly improved. In detail, it would be interesting to improve the following aspects:

* Introduction.

(1) It is a very short introduction and not very indicative of the state of the question. I believe that it is necessary to be somewhat more rigorous in the approaches and to expand the bibliographical references on the subject. I think it would be necessary to expand the studies that have been done on drought resistant varieties and that are cited worldwide. Therefore, I suggest that the following bibliographical references be analyzed:

  • Fischer RA Maurer R (1978) Drought resistance in spring wheat cultivars. I. Grain yield responses. Australian Journal of Agricultural Research 29, 897-912. https://doi.org/10.1071/AR9780897 (more than 1100 citations).
  • Blum, A. (2005). Drought resistance, water-use efficiency, and yield potential - are they compatible, dissonant, or mutually exclusive? Australian Journal of Agricultural Research, 56(11), 1159-1168. doi:10.1071/AR05069 (794 citations in Scopus database)
  • Maestre, F. T., Quero, J. L., Gotelli, N. J., Escudero, A., Ochoa, V., Delgado-Baquerizo, M., . . . Zaady, E. (2012). Plant species richness and ecosystem multifunctionality in global drylands. Science, 335(6065), 214-218. doi:10.1126/science.1215442 (642 citations in Scopus database).

Reply:

We analyzed and extracted relevant supportive information in the references above to improve the state of the question in the revision of the Introduction in our revised manuscript (lines 38-39, 80 -89, & 94-96 in clean version of revised manuscript).

(2) Also, how is your article in the framework of the Sustainable Development Goals (SDGs), and has there been any impulse to the issue you raise in your article from goals 13 and 15, for example? I am enclosing some of the most cited works worldwide for this purpose:

  • Fu, B., Stafford-Smith, M., Wang, Y., Wu, B., Yu, X., Lv, N., . . . Chen, X. (2021). The global-DEP conceptual framework — research on dryland ecosystems to promote sustainability. Current Opinion in Environmental Sustainability, 48, 17-28.  doi:10.1016/j.cosust.2020.08.009
  • Leach, M., Rockström, J., Raskin, P., Scoones, I., Stirling, A. C., Smith, A., . . . Olsson, P. (2012). Transforming innovation for sustainability. Ecology and Society, 17(2) doi:10.5751/ES-04933-170211
  • Stringer, L. C., Reed, M. S., Fleskens, L., Thomas, R. J., Le, Q. B., & Lala-Pritchard, T. (2017). A new dryland development paradigm grounded in empirical analysis of dryland systems science. Land Degradation and Development, 28(7), 1952-1961. doi:10.1002/ldr.2716

Reply:

We analyzed and incorporated relevant supportive information in the references above in our clarification of the relationship between the article and the sustainable development goals (SDGs) in the Introduction of our revised manuscript (lines 39-46, & 116 -121 in clean version of revised manuscript).

* Methodology section. It would be desirable to illustrate the area in which the study was developed. The authors could consider including a map showing the territory where the study was developed. As a reader of your work, I think this would be a good idea.

Reply:

A new map of the territory and relevant landcover was added in the Methods section of the manuscript (new Figure 1 in clean version of  revised manuscript).

* Finally, the article seems to me to be very complete and rigorous, but I think that the authors could exploit the conclusions a little more and not be so brief in this section of the article. I think it would be necessary to present a reflection on the need for governments to take note of this type of article, since it is the public administration that has to oversee this change. In this sense, it is curious to note that there is no reflection on the SDGs and the impetus given to the sustainability of the planet and the need to stop climate change.

Reply:

We have revised the Conclusions section (lines 440-448 in clean version of revised manuscript); Vegetation succession may lead to shifts in ecosystem composition in favor of more drought tolerant species, such as A. ordosica, in the near-to-immediate future. This may result in a reduction in biodiversity and ecosystem functioning, and enhanced vulnerability to global climate change and human disturbance in drylands. Dryland social-ecological systems are especially sensitive to rapid rates of physical and social change, such as associated with climate change and urbanization. Our findings demonstrate the need to strengthen dryland ecosystem management methods for sustainable livelihoods and to advance progress towards the implementation of 2030 SDGs for drylands.

Reviewer 4 Report

Reviews of Manuscript No.: forests-1382254

Title: Dynamic changes in plant resource use efficiencies and their primary influence mechanisms in a typical desert shrub community

Author(s): Yan Jiang, Yun Tian, Tianshan Zha, Xin Jia, Charles P.-A. Bourque, Peng Liu, Chuan Jin, Xiaoyan Jiang, Xinhao Li, Ningning Wei, and Shengjie Gao

Overall conclusions and recommendations:

This manuscript tackles an original question regarding plant resource use efficiencies. This study focuses on two widely distributed plant species (Artemisia ordosica and Leymus secalinus) in a typical desert shrub community, and measures their leaf photosynthesis, specific leaf area (SLA), leaf nitrogen concentration (i.e., Nmass + Narea), and several site-related abiotic factors in an entire growing season, from May to September 2019. Based on these measurements, authors calculated and analysed the water, light and nitrogen use efficiencies of the two plants. The key finding is that the RUEs for A. ordosica were mostly greater as compared to that for L. secalinus, but less sensitive to drought, suggesting that the shrub was more conservative in resource uses than the grass and was better able to adjust to arid conditions.

The study is within the scope of the journal Forest, and its topic and conclusions could attract the interest of readers working in areas of climate change adaption and land use/cover management. This manuscript has a clear structure, the quality of presentation is acceptable. The data measurement and analysis support the hypothesis. It has potential to be a good reference. I give a high recommendation of this manuscript.

Author Response

Dear Reviewer,

Thank you very much for the careful review of our manuscript (Forests-1382254), entitled “Dynamic changes in plant resource use efficiencies and their primary influence mechanisms in a typical desert shrub community.” We are grateful to you for an overall positive evaluation on our study!

Yours sincerely,

Dr. Yun Tian

Round 2

Reviewer 2 Report

Dear Authors,

Thank you for taking my comments into account.

Regards

Reviewer 3 Report

Dear Authors, 

Congratulations on your work. Thank you for taking my suggestions into account. I have recommended that it be accepted in its current form.

Regards, 

Reviewer.